# High-Level Production of Bacteriotoxic Phospholipase A1 in Bacterial Host *Pseudomonas fluorescens* via ABC Transporter-Mediated Secretion and Inducible Expression

**DOI:** 10.3390/microorganisms8020239

**Published:** 2020-02-11

**Authors:** Jiyeon Park, Gyeong Tae Eom, Joon Young Oh, Ji Hyun Park, Sun Chang Kim, Jae Kwang Song, Jung Hoon Ahn

**Affiliations:** 1Korea Science Academy of Korea Advanced Institute of Science and Technology, Busan 47162, Korea; jyp131@kaist.ac.kr; 2Intelligent Synthetic Biology Center, 291 Daehak-ro, Yuseong-gu, Daejeon 305-701, Korea; sunkim@kaist.ac.kr; 3Research Center for Bio-based Chemistry, Korea Research Institute of Chemical Technology (KRICT) 1, Ulsan 44429, Korea; eomgt@krict.re.kr; 4Research Center for Bio-based Chemistry, Korea Research Institute of Chemical Technology (KRICT), Daejeon 34114, Korea; jyoh@krict.re.kr (J.Y.O.); jhpark2@krict.re.kr (J.H.P.); ajee0414@gmail.com (J.K.S.); 5Department of Biological Sciences, Korea Advanced Institute of Science and Technology, Daejeon 34141, Korea

**Keywords:** phospholipase A1 (PLA1), PlaA, ABC transporter, *Pseudomonas fluorescens*, protein secretion

## Abstract

Bacterial phospholipase A1 (PLA1) is used in various industrial fields because it can catalyze the hydrolysis, esterification, and transesterification of phospholipids to their functional derivatives. It also has a role in the degumming process of crude plant oils. However, bacterial expression of the foreign PLA1-encoding gene was generally hampered because intracellularly expressed PLA1 is inherently toxic and damages the phospholipid membrane. In this study, we report that secretion-based production of recombinant PlaA, a bacterial PLA1 gene, or co-expression of PlaS, an accessory gene, minimizes this harmful effect. We were able to achieve high-level PlaA production via secretion-based protein production. Here, TliD/TliE/TliF, an ABC transporter complex of *Pseudomonas fluorescens* SIK-W1, was used to secrete recombinant proteins to the extracellular medium. In order to control the protein expression with induction, a new strain of *P. fluorescens*, which had the *lac* operon repressor gene *lacI*, was constructed and named ZYAI strain. The bacteriotoxic PlaA protein was successfully produced in a bacterial host, with help from ABC transporter-mediated secretion, induction-controlled protein expression, and fermentation. The final protein product is capable of degumming oil efficiently, signifying its application potential.

## 1. Introduction 

Phospholipase A1 (PLA1) (EC 3.1.1.32) hydrolyzes the sn-1 acyl ester bonds in phosphoglycerides, forming fatty acids and lysophospholipids [1]. Several PLA1-encoding genes from microorganisms, such as *Serratia* sp. MK1 [2], *Aspergillus oryzae* [3], *Escherichia coli* [4], and *Serratia liquefaciens* [5], have been cloned and expressed. In the last decade, there has been a great interest in the commercial use of PLA1 for food, nutraceuticals, pharmaceuticals, and oil degumming [6,7]. Lysophospholipids, produced by the hydrolytic action of PLA1, can be used as surfactants and functional ingredients in foods, cosmetics, and pharmaceutical products [8]. The PLA1-mediated enzymatic degumming process, which converts nonhydratable phospholipids to hydratable lysophospholipids in renewable oils, has been suggested as an efficient and environment-friendly alternative to conventional chemical processes [6,9,10].

Bacterial PLA1 is not well understood, and no crystal structure exists for any true PLA1 [1]. Minimal information is available on bacterial PLA1 due to the lack of any efficient method to clone, express, purify, and characterize the enzyme via large-scale cultivation. Bacteria use PLA1 as a virulence factor to invade the host cell, but it can also destroy their own membrane [11] and bacterial cell physiology [2,12]. Therefore, devising a method to mass-produce a bacterial PLA1 enzyme efficiently has significant importance in the industrial field. We chose *plaA*, a PLA1 gene from the Gram-negative bacterium *Serratia* sp. MK1, as the model bacterial PLA1 enzyme for our experiments. 

According to a previous study, PlaA can be expressed in *E. coli,* but most of the recombinant PlaA accumulates in the cytoplasm of the *E. coli* cells [2]. Since PlaA is toxic to the expression host cells because it hydrolyzes the cell membrane’s phospholipid, this study hypothesized that toxic PlaA could be produced more effectively by secreting it to the extracellular medium. To this end, PlaA was introduced into *Pseudomonas fluorescens*, which carries an ATP-binding cassette (ABC) transporter capable of secreting a variety of recombinant proteins, provided that they are fused with an ABC transporter signal sequence. In addition, *P. fluorescens* has several advantageous features as an expression host for recombinant protein production, such as its safety [13,14], adequacy for high cell density culture [15,16], and its exploitable export system [17]. Specifically, it has an ABC transporter complex TliDEF that functions in hydrolase enzyme secretion [18] and can be supplemented with additional copies of *tliDEF* genes to enhance recombinant protein secretion [17]. The *P. fluorescens* ABC transporter, TliDEF, is composed of three different protein multimers: TliD, TliE, and TliF, wherein TliD is an ATP binding cassette (ABC), TliE a membrane fusion protein (MFP), and TliF an outer membrane factor (OMF) [18]. TliDEF recognizes a C-terminal signal polypeptide sequence, LARD3, which exists in the cargo proteins. By fusing LARD3 with the target proteins, TliDEF can be used to export the recombinant target proteins. Using the TliDEF ABC transporter, we expected to translocate PlaA immediately after its initial synthesis in the cytoplasm. This way, PlaA would no longer exert any significant influence on the cellular membrane biogenesis and physiology. In addition, the extracellular secretion-based production simplifies the purification steps because it removes the need for the energy-costly cell lysis step, and then the culture supernatant is made free of the cytosolic contaminant proteins.

## 2. Materials and Methods 

### 2.1. Bacterial Strains, Growth Conditions

All genetic manipulations were performed with *Escherichia coli* XL1-Blue. The expression of each vector was conducted using *E. coli* XL1-Blue, *Pseudomonas fluorscens*
*ΔtliA*
*ΔprtA* (*P. fluorescens*
*Δtp*) [19], and *P. fluorescens*
*ΔtliA*
*ΔprtA algD::lacZYAI* (*P. fluorescens* ZYAI) as the host strain. While *E. coli* XL1-Blue was cultured in a LB medium with 60 μg/mL kanamycin at 37 °C, *P. fluorscens* was cultured in a LB or M9 medium with 60 μg/mL kanamycin at 25 °C. We used a slightly modified recipe for the M9 medium, which consists of an ordinary concentration of M9 salts, 2 mM MgSO_4_, 5% glycerol, trace elements (TE), and 0.1 mM CaCl_2_. TE was diluted from a 100 × TE stock solution, which was prepared as follows: 13.4 mM EDTA, 3.1 mM FeCl_3_, 0.62 mM ZnCl_2_, 76 μM CuCl_2_, 42 μM CoCl_2_, 162 μM H_3_BO_3_, and 8.1 μM MnCl_2_. The growth of *E. coli* harboring different plasmids was calculated from measured *A*_600_ values based on logistic cell growth curves, where the initial absorbance of each culture tube was set to be the same value by diluting the overnight culture.

### 2.2. Construction of Plasmids

The *plaA* and *plaS* genes (GenBank U37262) were amplified from *Serratia* sp. MK1 (KCTC 2865 from Korea Collection for Type Cultures) genomic DNA [20] and inserted into pDART, which carries the ABC transporter genes, as well as a C-terminal signal sequence gene, fused to the multiple cloning sites of the original vector pDSK519 [17]. We added codons of a His-tag at the 3’ end position of the *plaA* gene via a tagged PCR primer for detection, and a ribosomal binding site was inserted in the 5’ end. The pDART vector has genes coding for the *tliD/tliE/tliF* and a lipase ABC transporter recognition domain 3 (LARD3). To make pDART-PlaA (pABC/PlaA) and pDART-PlaA/PlaS (pABC/PlaA/PlaS), the sequence coding for His-tagged PlaA cleaved with XbaI and KpnI was inserted into pDART (pABC). The PCR-amplified *plaS* was inserted into pABC/PlaA, along with its original ribosomal binding site (RBS), using the In-Fusion cloning kit (Takara, Japan). As a control group, pPlaA was constructed by removing the ABC transporter genes from pABC/PlaA using BsrGI and ligation (no ABC transporter genes but C-terminally fused signal sequence is retained). As another control experiment, amplified *plaA* was cleaved with SphI and KpnI and then ligated with pDSK519, resulting in pDSK-PlaA (no ABC transporter genes, no signal sequence). From here, the *plaS* was amplified, along with its original RBS, cleaved with XbaI and SacI, and then ligated with pDSK-PlaA, resulting in pDSK-PlaA/PlaS. The transformed colonies were isolated on LB plates containing 30 μg/mL kanamycin and lecithin (L-α-phosphatidylcholine from egg yolk, Merck, Germany) to check PlaA activity. We followed the standard protocols for plasmid isolation, restriction endonuclease digestion, ligation, polymerase chain reaction (PCR), and gel electrophoresis procedures [21]. *E. coli* transformation was performed using conventional heat shock methods, and *P. fluorescens* transformation was performed via electroporation at 2.5 kV, 125 Ω, and 50 μF with electrocompetent cells [17] or via conjugation [22].

### 2.3. Construction of P. fluorescens ZYAI and P. fluorescens AlgD::LacI

The *lacI^q^* gene was amplified from pBB528 [23] using *lacI* primers such that the promoter and transcription terminator of *lacI^q^* were contained in the PCR fragment. The *algD* upstream part (*algD1*) was also amplified using *algD1* primers such that the stop codon and the transcription terminator for the *algD* operon were contained in the PCR fragment. The *algD* downstream part (*algD2*) was amplified using *algD2* primers. These three PCR fragments and pK19 *mobsacB* [24] were combined to make pK-lacI using an In-Fusion cloning kit. The *lac* operon (including a promoter, an operator, *lacZ*, *lacY*, and *lacA*) was amplified, using lacZYA primers, and inserted into pK-lacI using restriction enzyme sites HindIII and PstI. The resulting pK-lacI and pK-lacIZYA plasmid were transformed into *E. coli* S17-1 for conjugal transfer into *P. fluorescens*
*Δtp*. The *P. fluorescens Δtp* with the inserted *lacI* or *lac* operon was screened as previously reported [19]. Single recombinants were screened on M9 containing 0.6% lactose to check the activity of the *lac* operon. The colonies of the single recombinants were grown in 10% sucrose, and double recombinants were then screened on 10% sucrose-LB plates. The sequences of all primers used in this study are presented in Table 1.

### 2.4. Analyses of PlaA Expression and Fermentation

Recombinant cells were grown in the LB or M9 medium supplemented with 60 μg/mL kanamycin. When *P. fluorescens* ZYAI reached 0.8 absorbance (path length 1 cm) at 600 nm (*A*_600_), 1 mM (final concentration) of isopropyl-β-D-thiogalactopyranoside (IPTG) was added to the culture in order to induce PlaA expression. To separate the supernatant and cell pellet, the culture broth was centrifuged at 18,000 rcf for 10 min. The proteins of the cell pellet and the supernatant were analyzed using sodium dodecylsulfate-polyacrylamide gel electrophoresis (SDS-PAGE) in 10% polyacrylamide gels, following Laemmli’s method [25]. The proteins were transferred onto a nitrocellulose membrane (Amersham, UK) for western blotting, performed as previously described [25] using anti-His primary antibody (Qiagen, Germany) and anti-mouse IgG secondary antibody chemiluminescence system (Advansta, San Jose, CA, USA). 

To examine the possibility of high-level secretion-based PlaA production by the fed-batch fermentation, recombinant *P. fluorescens* ZYAI harboring pABC/PlaA was cultivated in a 500 mL flask containing 200 mL LB with 60 μg/mL kanamycin at 30 °C for 24 h. The seed culture was inoculated in a 2 L M9 medium with 60 μg/mL kanamycin. Batch fermentation was performed at 30 °C in a 5 L jar fermenter (New Brunswick BioFlo 310, Eppendorf, Germany). After batch fermentation, IPTG was added at a final concentration of 1 mM, and 50% glycerol solution with 60 μg/mL kanamycin and trace metal solution was fed into the culture at a feeding rate of 5 mL/h. Fed-batch fermentation was performed at 25 °C. The pH was controlled at 7.4 by adding 15% aqueous ammonia, and the dissolved oxygen level was constantly adjusted to 30% by controlling the agitation speed, airflow, and supplemental pure oxygen flow during batch and fed-batch fermentation.

### 2.5. Measurement of Secretory PlaA Activity 

PlaA activity was detected by directly cultivating cells on lecithin agar plates, which were prepared by adding 1.5% phosphatidylcholine (Amresco, Solon, OH, USA), 0.5% taurocholic acid (Merck, Germany), 10 mM CaCl_2_, and antibiotics to autoclaved 1.5% LB agar. After the colony was incubated at 25 °C for four days, the phospholipase activity zone (Pz) was measured. As described by Price et al. [26,27], Pz was calculated by dividing colony diameter with the colony diameter plus opaque and transparent diameter around the colony. When Pz = 1, PlaA activity is considered negative and, as PlaA activity increases, Pz value approaches zero. We quantified PlaA activities on lecithin agar plates using the 1-Pz scoring system. Experiments were carried out on three separate occasions using a multi-gauge program.

The relative activity of the secreted PlaA was measured using N-((6-(2, 4-DNP) amino) hexanoyl) 1-(BODIPY FL C5)-2-hexyl-*sn*-glycero-3-phophoethanolamin (PED-A1) (Invitrogen, USA) as a fluorogenic phospholipase A1 substrate. The PED-A1 solution consists of 45 nM PED-A1, 10 mM Tris-HCl (pH 8.0), 100 mM NaCl, and 10 mM CaCl_2_. A 90 μL PED-A1 solution was incubated with 10 μL supernatant of the culture medium in a 96-well microplate [28]. A Tecan-Genios-Pro multimode microplate reader was used to measure fluorescence intensity, and Magellan software was used to analyze the measurements. The fluorescence intensity was determined with an incident excitation light of wavelength of 485 nm, and emission was detected at wavelength of 538 nm. For the absolute estimation of PlaA activity in unit, the pH-stat method was employed using an automatic pH titrator (Metrohm Tiamo, Switzerland). The lecithin substrate containing 20 mM lecithin, 6.4 mM CaCl_2_, and 3.2 mM sodium deoxycholate was homogenized for 10 min. Then, the rate of reaction was monitored by titrating with 10 mM NaOH at pH 8.0 and 40 °C for 3 min [29]. One unit (U) was defined as the release of 1 μM of fatty acid per min under experimental conditions. 

### 2.6. Degumming of Crude Plant Oil and Lecithin Hydrolysis

The fermentation broth containing the secreted PlaA was diafiltrated using a tangential flow filtration membrane with a molecular weight cut-off of 10 kDa in 50 mM Tris-HCl (pH 8.0). For the PlaA-catalyzed degumming process, the 5 mL of crude sesame oil was heated to 60 °C in a water bath for 1 h and cooled down to 40 °C. The PlaA solution was added to crude sesame oil and vortexed for 1 min. The degumming reaction was conducted by shaking at 300 rpm and 40 °C for 24 h as previously reported [12]. 

The substrate solution for lecithin hydrolysis consisted of 10% (*w/v*) lecithin, 600 mM sodium chloride, 20 mM calcium chloride, and 1 mM sodium taurocholate. The substrate solution pH was adjusted to 8.0 using a 5 N sodium hydroxide solution. The hydrolysis reaction of lecithin was started by adding an appropriate amount of PlaA solution to 30 mL of the substrate solution and incubating at 40 °C while being stirred with a magnetic stirrer. After the reaction was terminated, the amount of lecithin converted to lysolecithin was measured by titrating the released fatty acids with a 0.1 N NaOH solution to raise back to pH 8.0. 

## 3. Results 

### 3.1. PlaA Expression in E. coli 

*Serratia* species secrete PlaA naturally with the help of *plaS,* which is juxtaposed with *plaA* as an operon [2]. However, PlaA was localized inside *E. coli* cells despite being co-expressed with PlaS [2]. Furthermore, only a marginal amount of recombinant PlaA is obtained from liter-scale *E. coli* cultures due to substantial inhibition of cell growth and protein biosynthesis [12]. In this study, we intended to produce PlaA efficiently in a bacterial host by secreting the PlaA via the functionally reconstituted ABC transporter system, TliDEF, in a heterologous bacterial strain (Figure 1). We needed to understand how the two different functional elements, PlaS and the ABC transporter, interplay in protein secretion. For this purpose, various plasmids expressing PlaA attached to LARD3 and PlaS were constructed. 

All plasmids were constructed using pDSK519 [30], a broad host range vector, for expression in different hosts (Figure 1A). We first analyzed the constructed plasmids in *E. coli* to check if the plasmids were constructed properly. *E. coli* harboring different plasmids was inoculated using toothpicks onto lecithin plates and incubated (Figure 2A). PlaA activity can be detected on a lecithin plate by observing the conversion of lecithin into visually detectable substances, which form a double-layered halo around each colony [31]. Within each halo, an opaque halo, composed of glycerophosphocholin and fatty acid precipitation, was formed in the inner region (near the colony), with a transparent halo, consisting of water-soluble lysolecithin, in the outermost region. In our experiments, *E. coli* harboring *plaA* showed double-halos, which included both opaque and transparent halos. The halo size was increased further by supplementing the cells with *plaS*. The PlaA activity level was quantified by Pz value and activity of colonies harboring both *plaA* and *plaS* was the highest (Figure 2B). Interestingly, the colonies harboring *plaA* without *plaS* were somewhat translucent on the LB plate (Figure 2C), and they exhibited retarded cell growth (Figure 2D). We believe these symptoms indicate that the expression of PlaA without PlaS in *E. coli* is toxic to the host cell. The toxic effect was alleviated by co-expression of PlaS, making the colonies on the agar plate opaque again and rescuing the growth rate. Next, we scrutinized the liquid culture of *E. coli* supplemented with the genes for the ABC transporter, but there was no detectable amount of secreted PlaA in the culture supernatant of the cells harboring *plaA* alone or both *plaA* and *plaS* (Figure 2E). Perhaps, we could not detect PlaA in the liquid culture supernatant because of the suppressive regulation of gene expression in liquid culture, in contrast to agglomerated cell colonies on the solid agar plate. *Serratia* sp. MK1 was also tested for PlaA production with plasmids shown in Figure 1A, but there was no additional secretory PlaA production compared to the wild type strain. In any case, these results indicate that *E. coli* and *Serratia* sp. are not a viable expression host for PlaA expression. Therefore, we decided to test whether *P. fluorescens* could be a viable replacement.

### 3.2. Construction of P. fluorescens ZYAI 

We hypothesized that *P. fluorescens* could be an appropriate host for *plaA* expression since *plaA* is closely related to *Serratia* sp. MK1, the origin of *plaA*, and both of them are categorized in the same class of γ-proteobacteria. Furthermore, it is the natural host of the ABC transporter we are using and is proven capable of secreting many different types of recombinant proteins using the ABC transporter. The constructed plasmids (Figure 1A) were transformed into *P. fluorescens*
*ΔtliA*
*ΔprtA* (hereafter *Δtp)* via electroporation*. P. fluorescens*
*Δtp*, a knockout mutant of *P. fluorescens* SIK-W1 with *tliA* (lipase) and *prtA* (protease) genes knocked out, exhibits a superior level of detectable recombinant proteins in an extracellular medium when used as an expression host [19]. Although we were able to isolate *E. coli* colonies expressing PlaA, we failed to isolate *P. fluorescens*
*Δtp* transformed with pABC/PlaA which expresses *plaA* without *plaS*, as these cells did not make any colony. This is because *E. coli* has an expression control mechanism and *P. fluorescens* does not. On the agar plate, where no inducing agent is present, *E. coli* can grow as it expresses the LacI repressor from its genomic *lacI* gene, and this represses the *lac* promoter of our plasmid, enabling colony growth. It seems that uncontrolled PlaA expression without PlaS is too toxic for the cells to produce colonies. On the other hand, *P. fluorescens*
*Δtp* transformed with pABC/PlaA/PlaS made a few colonies because PlaS allowed *P. fluorescens* growth by mitigating PlaA toxicity in the cell. For controllable PlaA expression, we constructed a new knock-in mutant of *P. fluorescens*
*Δtp* using the *E. coli lac* operon system.

The entire *lac* operon from *E. coli,* including the *lacI* gene, was inserted into the chromosome of *P. fluorescens*
*Δtp*. The *lac* operon (*lacZ, lacY, lacA, and lacI*) was inserted into the first gene *algD* of the *alg* operon to form the knock-in mutant *P. fluorescens* ZYAI (Appendix A). The *alg* operon [32,33] was selected as the knock-in insertion site because biosynthesized alginate, the metabolic product of *alg* operon genes, is assembled as a biofilm matrix in vivo [34,35] and is not beneficial for the suspension culture in liquid media. 

The *lac* operon knocked-in *P. fluorescens* ZYAI could feed on lactose as its sole carbon source, and it made blue colonies using IPTG and X-gal, while *P. fluorescens*
*Δtp* and another mutant *P. fluorescens algD*::*lacI* cannot digest lactose (Appendix A). *P. fluorescens algD*::*lacI* was not induced by IPTG for gene expression under the *lac* promoter (Appendix A). This was caused by a lack of *lacY* and *lacZ*, which transport lactose and convert it to allolactose, the natural inducer [36]. The PlaS was essential when toxic PlaA was expressed in *P. fluorescens*
*Δtp*; however, PlaA could be expressed without PlaS in *P. fluorescens* ZYAI. All of the *P. fluorescens* ZYAI colonies harboring *plaA* were opaque and grew well, unlike the *E. coli* colonies on LB plate as shown in Figure 2C,D, indicating that PlaA’s cytotoxic activity was downregulated by the repression control of the gene expression in *P. fluorescens* ZYAI. To determine the optimum IPTG dose, the PlaA expression level was examined at different IPTG concentrations and analyzed via western blot. At 1 mM IPTG, the highest PlaA secretion was observed. We also used M9 medium for the following experiments, not only because the PlaA expression level was higher in the M9 medium than in the LB (Appendix A), but also because PlaA takes up the majority of the M9 culture supernatant proteins with significantly fewer contaminant proteins.

### 3.3. Secretion of PlaA in P. fluorescens ZYAI

We examined the secretory production of PlaA in *P. fluorescens* ZYAI by introducing the plasmids described in Figure 1A. *P. fluorescens* ZYAI harboring pABC/PlaA formed colonies and showed an activity halo, including both the opaque and the transparent halos, while the colony harboring pABC/PlaA/PlaS showed a smaller activity halo (Figure 3A,B). To analyze PlaA secretion, *P. fluorescens* ZYAI harboring pABC/PlaA were cultured in M9 medium with IPTG induction, and the secretion was compared with *P. fluorescens* ZYAI harboring pABC/PlaA/PlaS (Figure 3C). Much more PlaA was secreted by the ABC transporter when *plaA* was expressed without *plaS,* while only a trace activity was measured in cells containing both *plaA* and *plaS* (Figure 3D). Moreover, *P. fluorescens* ZYAI expressing *plaA* secreted three times more PlaA than the noninducible *P. fluorescens*
*Δtp* supplemented with both *plaA* and *plaS* (Figure 3E). *P. fluorescens* ZYAI harboring pABC/PlaA/PlaS showed the lowest PlaA activity in the culture supernatant. The induction-controlled PlaA expression in *P. fluorescens* ZYAI exhibited much better protein secretion than *P. fluorescens*
*Δtp* with co-expressed *plaS*. 

It was evident that the ABC transporter-mediated secretion of PlaA is crucial to the high-level secretion of PlaA in the inducible *P. fluorescens* ZYAI. As a control experiment, the ABC transporter’s role in PlaA production was tested by checking the level of culture supernatant without the ABC transporter or LARD3, the C-terminal signal sequence. Without the ABC transporter in the plasmid, PlaA was only minimally localized to the extracellular medium, perhaps by the ABC transporter expressed from the single copy of the genomic *tliDEF* gene, which is not overexpressed (Appendix A). The wild-type PlaA, which lacks the LARD3 signal sequence, was also tested in *P. fluorescens*, but PlaA lacking LARD3 was not localized to the culture supernatant (Appendix A). These results confirmed that PlaA was secreted only by the conjugated LARD3 and the supplemented ABC transporter in *P. fluorescens*.

### 3.4. Production of PlaA in Fermenter

In order to achieve high-level, medium-scale secretion-based PlaA production, an induced PlaA expression in *P. fluorescens* ZYAI containing pABC/PlaA was carried out in a fermenter using a two-phase protocol consisting of an M9-glycerol batch at 30 °C and glycerol fed-batch at 25 °C (Figure 4). To proceed to the protein production phase after the 45 h batch culture, 1 mM IPTG (final concentration) was added to induce the PlaA production. Even though the cell biomass was in a stationary state at *A*_600_ around 16 while the culture was being fed constantly with the supplemental glycerol solution, the PlaA activity in the culture supernatant significantly increased up to 25 units/mL during 51 h glycerol fed-batch phase (Figure 4A). We also confirmed that the presence of 45.7 kDa PlaA in the culture supernatant via western blot (Figure 4B). The PlaA activity of the fed-batched culture supernatant at 96 h was estimated by pH-stat assay, and the result showed that it was approximately 50-fold higher than that of the 5 mL test tube culture supernatant (Figure 4C). The secreted PlaA concentration, estimated from the band density of SDS-PAGE and the western blot, was 17 mg/L. The specific activity of PlaA measured by pH-stat was 1433 ± 139 U/mg, and the calculated turnover number (k_cat_) was 1091 s^−1^.

### 3.5. Use of Secreted PlaA in Degumming Crude Plant Oil and Hydrolysis of Lecithin 

The PlaA solution prepared from the fermentation was used for degumming crude sesame oil and hydrolysis of crude lecithin to test its suitability in bio-catalytic applications. First, crude sesame oil was incubated with 10 μg PlaA solution for 24 h at 40 °C to facilitate enzymatic degumming by PlaA. As shown in Figure 5A, cloudy crude oil was cleared as PlaA converted phospholipids to lysophospholipids. Next, we examined the hydrolysis of crude lecithin with PlaA. The increase of fatty acids released by the action of PlaA was directly proportional to the amount of enzyme used (Figure 5B), indicating that the recombinant PlaA successfully catalyzed lecithin hydrolysis. Time-course analysis showed that lecithin hydrolysis in 10% aqueous lecithin solution eventually reached a plateau after 4 h of reaction under the presence of the secreted PlaA solution (Figure 5C). At the plateau, there was about 630 μmole of lysolecithin, which corresponded to a conversion of 27.0% of the total lecithin. These results suggest that the PlaA prepared via the ABC transporter-mediated secretion in *P. fluorescens* could be a promising industrial solution to produce lysolecithin or degum crude renewable oils.

## 4. Discussion and Conclusions

In this report, we propose that ABC-transporter-based PlaA production can be a favorable alternative to the conventional cytoplasmic expression in bacteria. Since PLA1 is bacteriotoxic when accumulated in the cytoplasm, the mass production of foreign PLA1 with conventional methods was hampered by low cell growth and consequent global suppressive regulation of gene expression [12]. Previously, *plaA*, a bacterial PLA1 from *Serratia* sp. MK1 [2] and its engineered variants [37,38] have been expressed mainly in *E. coli*. Due to PlaA’s toxicity to cellular membranes, an effort to develop an efficient bacterial expression system has not been effective for bacterial PlaA production. We also attempted to express recombinant PlaA in *E. coli* or *Serratia* sp. MK1, but as the expressed PlaA accumulated in the cytoplasm, the bacteriotoxic activity of PlaA retarded cell growth (Figure 2C,D). When *P. fluorescens* strain equipped with an ABC protein exporter was used, the PlaA molecules that were initially synthesized in the cytoplasm could be translocated swiftly to the extracellular medium, where they can no longer exert any serious influence on cell viability (Figure 1B). We demonstrated that the ABC protein exporter-mediated secretion-based production in *P*. *fluorescens* can be a versatile tool for producing bacteriotoxic proteins that would be difficult to express otherwise. 

Gram-negative bacteria contain various secretion systems, ranging from the type I secretion system (T1SS) to the type VI secretion system (T6SS) [39]. The PlaA used in this study is originally secreted by the type III secretion system (T3SS), similar to *S. marcescens* PhlA [40] and Yersinia YplA [41]. These PLA1s are secreted through the flagella-related T3SS, which usually mediates the bacterial flagellar formation in *E. coli* and *Serratia* species [40,42,43,44,45,46]. PlaA, PhlA, and YplA have an N-terminal signal peptide that is about 19–23 residues of hydrophobic amino acids [47] and have been predicted to be secreted by the T3SS [48]. In our preliminary experiments, we tested PlaA secretion by native T3SS of *P. fluorescens* by transforming it with plasmids harboring unmodified versions (without LARD3 signal sequence) of *plaA* or *plaA*/*plaS* (Appendix A). The PlaA was not detected in the liquid culture supernatant even though PlaA activity was observable on the agar plate activity assay. The regulators and controllers of *P. fluorescens* T3SS did not seem to operate properly for the *Serratia* PlaA in liquid culture, perhaps because *Serratia* PlaA expression is delicately controlled by a flagellar regulator [5,43,49] or cysteine biosynthesis of *Serratia* [50]. However, PlaA could be secreted by C-terminal LARD3 signal sequence via *P. fluorescens* ABC transporter (T1SS) even though it is incompatible with *P. fluorescens* T3SS. 

The *plaA* is a toxic gene which can deteriorate membrane integrity by hydrolyzing phospholipids. The associated protein PlaS specializes in modulating or regulating PlaA activity. PlaS inhibits PlaA by interacting directly with PlaA, and it facilitates the protection of the cell interior [5,43,44,47]. Our results were consistent with this. In *E. coli*, PlaS restored normal colony physiology and cell growth rate (Figure 2C,D). It was similar in *P. fluorescens*
*Δtp*, which requires strict PlaS co-expression to survive on expressing PlaA. However, the PlaS was not necessary in the *lacI*-controlled *P. fluorescens* ZYAI strain. Furthermore, PlaS reduced the ABC transporter-mediated secretion of PlaA (Figure 3B). It seems that the intracellular interaction of PlaA and PlaS could interfere with the PlaA unfolding process, which happens during the T1SS-dependent secretion, resulting in a decreased secretion level (Figure 1B). It is noteworthy that heterodimer formation decreases the protein secretion across the membrane in many transport mechanisms [51,52]. Therefore, it is expected that the binding and subsequent heterodimer formation stabilizes the folded structure of the cargo proteins, resulting in a decreased secretion. The newly developed *lacI*-regulated *P. fluorescens* ZYAI can be used as an alternative to PlaS co-expression, where we can control the PlaA expression, constraining the damage to cell viability done by PlaA’s bacteriotoxic activity. The newly developed strain *P. fluorescens* ZYAI enabled us to use a convenient, controlled, and inducible expression of recombinant genes, permitting us to overproduce bacteriotoxic proteins such as PlaA.

In conclusion, we showed that bacteriotoxic PlaA can be produced and secreted using an ABC transporter in a bacterial host. The secreted enzyme can be put to use readily for degumming processes, such as the enzymatic conversion of lecithin to lysolecithin. The toxic protein PlaA, which previously proved difficult to express in bacterial cells, was produced successfully by fed-batch fermentation of the newly developed knock-in mutant strain *P. fluorescens* ZYAI, which has *lac* operon for controllable expression. The regulation of gene expression by *lacI* enabled reliable PlaA gene expression and allowed sustained PlaA production in the extracellular medium. The ABC transporter-mediated secretion in *P. fluorescens* ZYAI could be a promising alternative to the conventional intracellular protein production methods for the production of bacteriotoxic proteins such as PlaA.

## Figures and Tables

**Figure 1 microorganisms-08-00239-f001:**
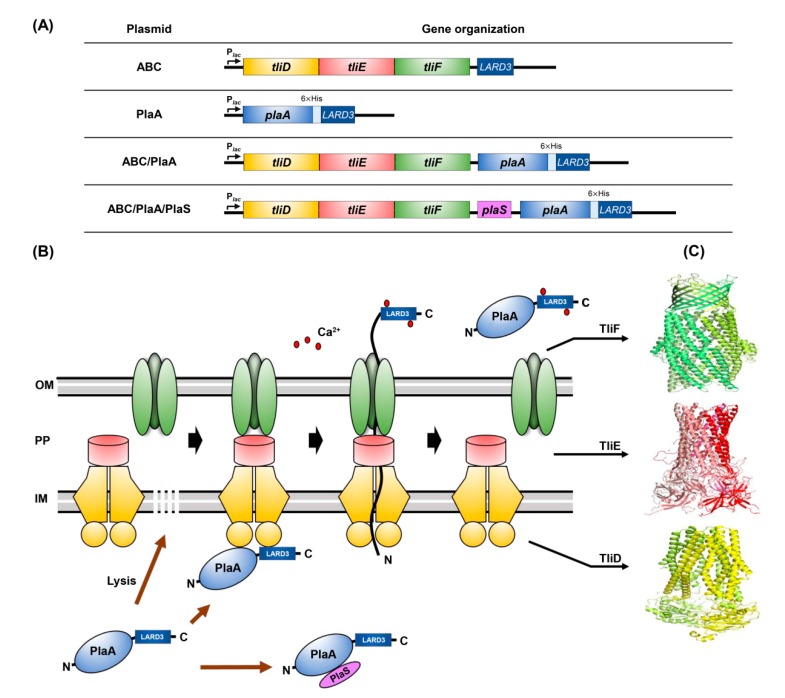
Construction of recombinant plasmids and the scheme of bacterial phospholipase A (PlaA) secretion by ABC transporter in *P. fluorescens* ZYAI. (**A**) Recombinant plasmids were constructed by incorporating *plaA* and *plaS* into pDART. All genes were under the control of the *E. coli lac* promoter. Each original ribosomal binding site is placed at the upstream positions of *plaA* and *plaS*. ABC: pDART; PlaA: pPlaA; ABC/PlaA: pDART-PlaA; ABC/PlaA/PlaS: pDART-PlaA/PlaS. (**B**) Schematic representation of PLA1 secretion by ABC transporter. The TliDEF are separated in the resting state. When the C-terminal of ABC transporter recognition domain 3 (LARD3) binds to TliD, the ABC transporter assembles. The PlaA is secreted into the extracellular medium through the ABC transporter. The extracellular calcium ions attach to certain parts of the signal sequence and pull the remaining protein out of the cell. If the PlaA is accumulated in the cell, it induces cell lysis. The PlaS reduces PlaA secretion with interaction of PlaA. (**C**) The three-dimenional (3D) structures of the TliDEF. The structures of dimeric TliD, hexameric TliE, and trimeric TliF were predicted by SWISS-MODEL. OM: outer membrane; PP: periplasm; IM: inner membrane.

**Figure 2 microorganisms-08-00239-f002:**
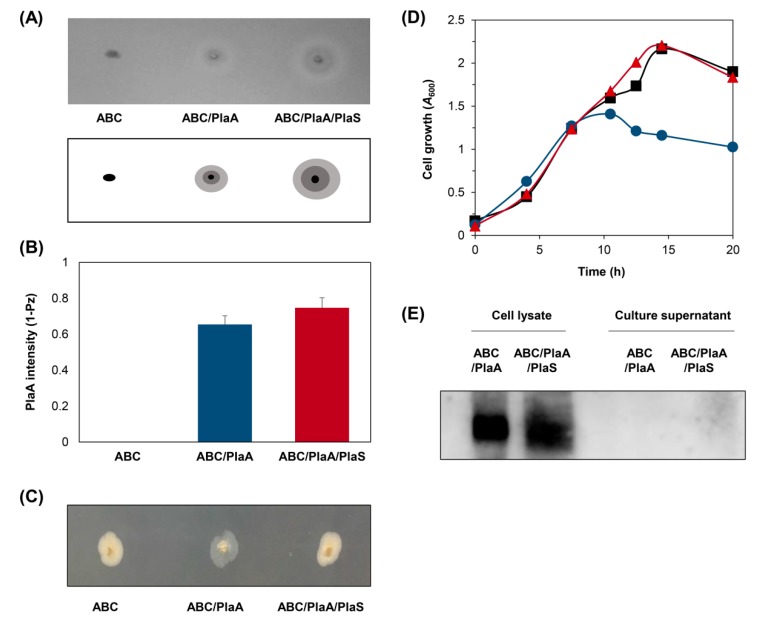
Phospholipase activity of recombinant PlaA in *E. coli*. (**A**) *E. coli* cells harboring each plasmid were grown on lecithin plates at 25 °C for two days. The schematic diagram illustrates zones of colony (black), an opaque halo formed by fatty acid precipitation (dark grey), and a transparent halo formed by water-soluble lysophospholipid (light grey). (**B**) PlaA activity was quantified by measuring halo zones around colonies on the lecithin plate. The Pz refers to colony diameter divided by colony diameter plus opaque and transparent zones diameter so 1 – Pz = 0 means no activity. (**C**) *E. coli* cells harboring each plasmid were grown on LB agar plate at 37 °C for four days. (**D**) Growth curves of *E. coli* harboring different plasmids, ABC (■), ABC/PlaA (●), ABC/PlaA/PlaS (▲). The cells were cultured in the LB media with 1 mM IPTG added at *A*_600_ = 0.8. (**E**) Western blot analysis of PlaA from the samples of cell lysates and culture supernatants. The plasmids shown in Figure 1A were used for comparison. Error bars represent the standard deviation from three independent experiments.

**Figure 3 microorganisms-08-00239-f003:**
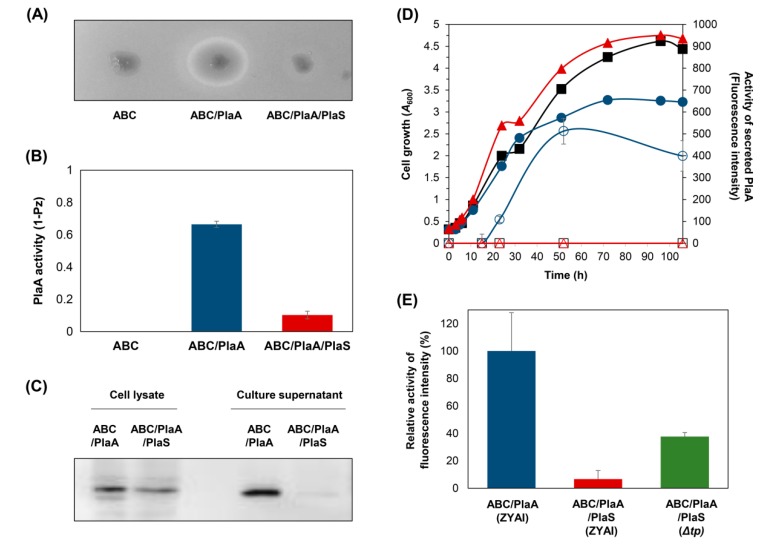
PlaA secretion by the ABC transporter system in *P. fluorescens*. (**A**) Each cell was grown on the lecithin plate at 25 °C for four days. (**B**) PlaA activity was quantified by measuring the diameter of the halo zones around colonies on the lecithin plate. (**C**) Western blot analysis of PlaA from the samples of cell lysates and culture supernatants. The cells were cultured in M9 medium for four days with 1 mM Isopropyl β-D-1-thiogalactopyranoside (IPTG) added at *A*_600_ = 0.8. (**D**) Growth curves of *P. fluorescens* ZYAI harboring different plasmids and activity of secreted PlaA, ABC (■ and □), ABC/PlaA (● and ○), ABC/PlaA/PlaS (▲ and △). The cells were cultured in M9 media with 1 mM IPTG added at *A*
_600_ = 0.8. The PlaA activity was measured using PED-A1, fluorogenic substrate specific for PlaA in the culture supernatant. (**E**) The relative activity of PlaA, ABC/PlaA, and ABC/PlaA/PlaS was measured using the fluorescence substrate PED-A1 with the activity of ABC/PlaA set at 100%. The plasmids shown in Figure 1 were used for the experiments. All experiments were performed with *P. fluorescens* ZYAI except ABC/PlaA/PlaS, the activity of which was compared in both *P. fluorescens*
*Δtp* and *P. fluorescens* ZYAI. Error bars represent the standard deviation from three independent experiments.

**Figure 4 microorganisms-08-00239-f004:**
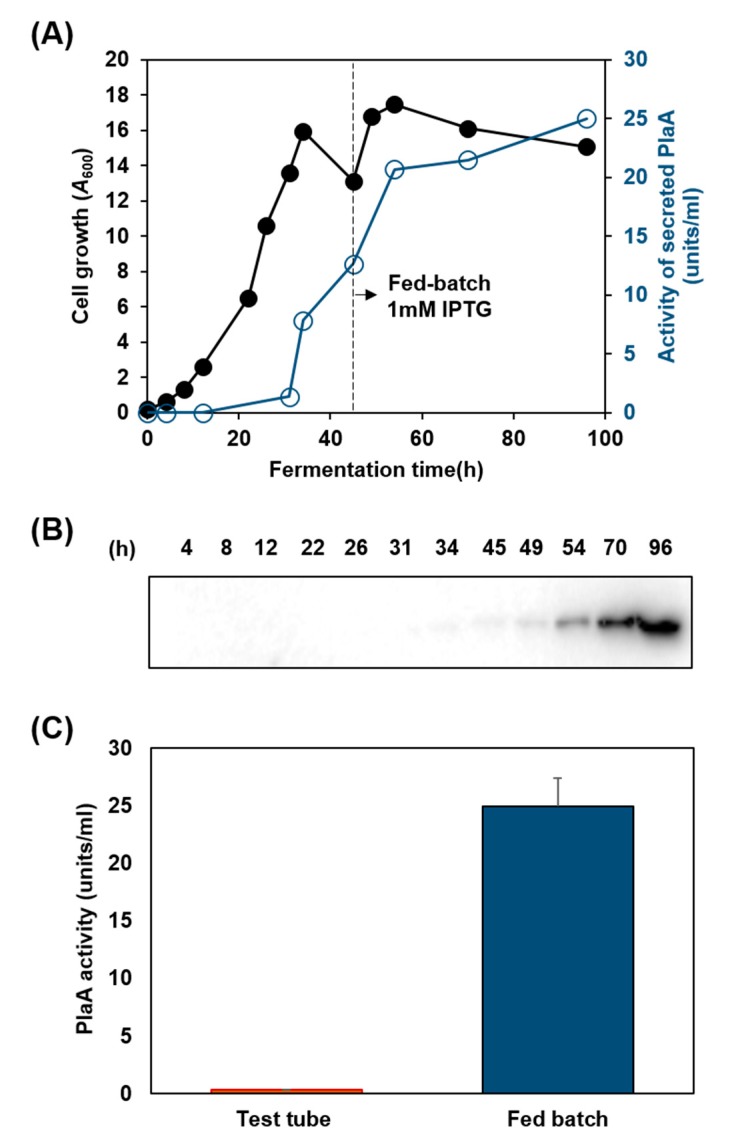
Fed-batch fermentation of *P. fluorescens* ZYAI harboring ABC/PlaA. The secretory production of PlaA by *P. fluorescens* ZYAI harboring ABC/PlaA was monitored with the samples collected from the fed-batch fermentation broth at different times. (**A**) Cell growth (●) was determined by measuring the optical density at a wavelength of 600 nm and the activity of secreted PlaA (○) was measured by pH-stat. (**B**) Western blot analysis was carried out with the culture supernatant using anti-His-tag. (**C**) The PlaA activity of 5 mL M9 medium and fed-batch fermentation in a 5 L fermenter was measured by pH-stat. Error bars represent the standard deviation from three independent experiments.

**Figure 5 microorganisms-08-00239-f005:**
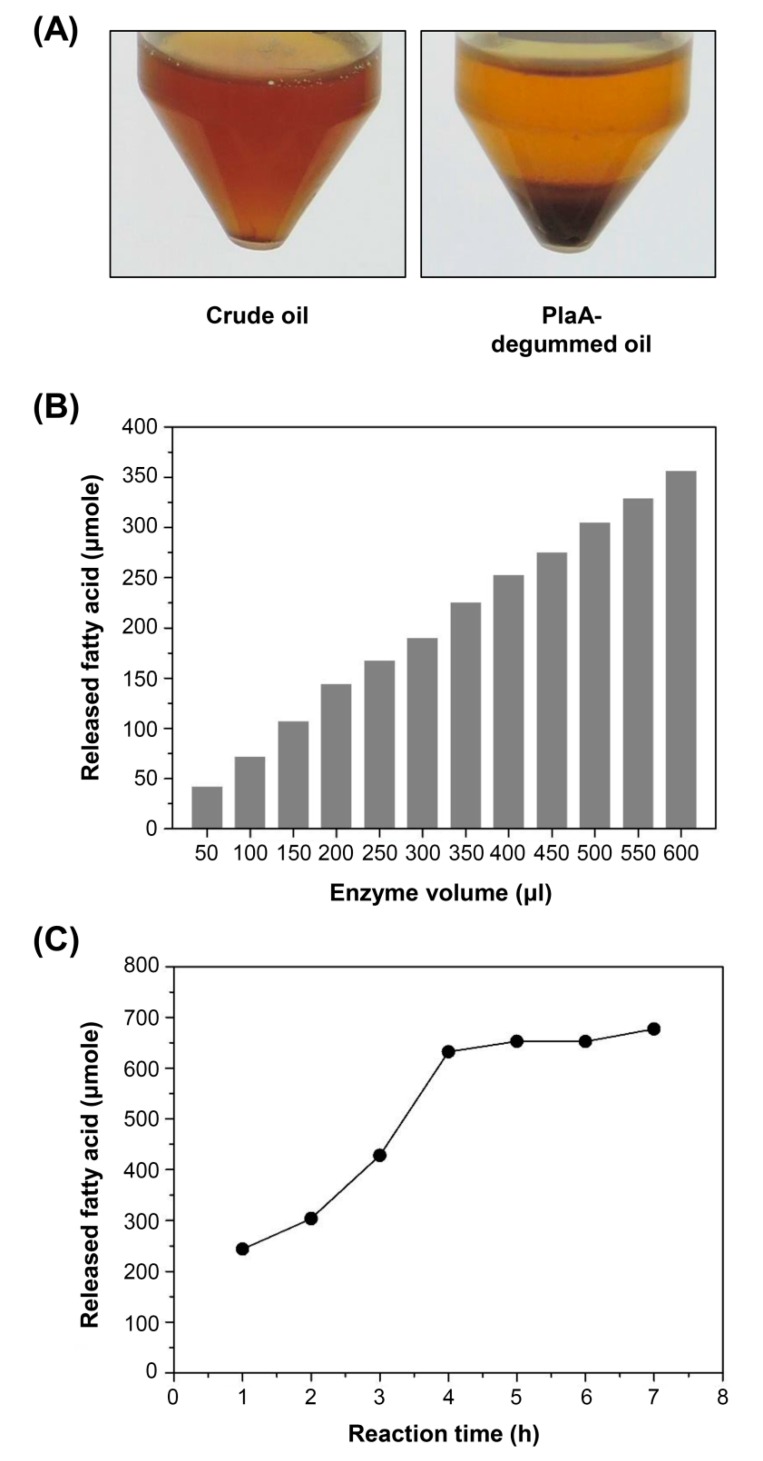
Degumming crude oil and lecithin hydrolysis using the culture supernatant of PlaA. (**A**) The crude sesame oil was incubated with 10 μg PlaA solution for 24 h at 40 °C. (**B**) Different amounts of PlaA solution were added to the substrate solution. After 2 h of incubation at 40 °C, the conversion of lecithin to lysolecithin was measured. (**C**) The conversion of lecithin to lysolecithin by the secreted PlaA was measured at various reaction times.

**Table 1 microorganisms-08-00239-t001:** List of primers used in this study.

Primer ID	Primer Sequence for PCR	Target Vector
lacI	F 5’ GCGGGGTTTTTTTTTAAGGCAGTTATTGGTCCCT 3’ R 5’ AAAAAGCCGCCAGCGGAACTGGCGGCGTGTGAAATTGTTATCCGCTC 3’	pK19
algD1	F 5’ GACGGCCAGTGAATTCCACGAAGTGGTCGGCGTAGA3’ R 5’ AAAAAAAAACCCCGCCGAAGCGGGGTCAGTCGACGCCGGCTTTCTTGC3’	pK19
algD2	F 5’ CGCTGGCGGCTTTTTTCCCAGTACTACATGCGCCC3’ R 5’ TGATTACGCCAAGCTGTTGAGCAGGGACGACACGT3’	pK19
lacZYA	F 5’ GGG AAGCTTGCGCAACGCAATTAATGTGAG 3’ R 5’ GGGCTGCAGGGTCAAAGAGGCATGATGCG 3’	pK-lacI
plaA-1	F 5’ TCTAGA ATGAGTATGTCTTTGAGTTTTAC 3’ R 5’ GGTACCGTGATGGTGATGGTGATGGGCATTGGCCATCGCCTCC 3’	pDART
plaS-1	F 5’ CAAGACAATGTCTAGGCCATGGGAGGCGATGGCC 3’ R 5’ ACATACTCATTCTAGCTCCTTGTCGTTACTGCTGTCCGTATTGCG 3’	pDART
plaA-2	F 5’ GCATGCCTAGCGACAAGGAGTCGGCATGA 3’ R 5 ’TCTAGATTAGTGATGGTGATGGTGATGGGCATTG GCCATCGCCTC 3’	pDSK519
plaS-2	F 5’ TCTAGAGCCATGGGAGGCGATGGC 3’ R 5’ AGCTCTTACTGCTGTCCGTATTGCG 3’	pDSK519

F denotes forward primer. R denotes reverse primer.

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
