# Peer review of "High-Level Production of Bacteriotoxic Phospholipase A1 in Bacterial Host Pseudomonas fluorescens via ABC Transporter-Mediated Secretion and Inducible Expression"

_microorganisms, 2020, doi:10.3390/microorganisms8020239_

Round 1
Reviewer 1 Report
The authors aimed to produce a new bacterial strain for the higher expression levels of PlaA, a bacterial enzyme with phospholipase A1 activity that can be harmful to the phospholipid-containing membrane. They achieved their goals by (1) having the ATP-binding cassette (ABC) transporter complex (TliD/TliE/TliF) on the bacterial membrane (2) incorporating a lacI repressor gene for induction-controlled expression system. Overall, the strategy is novel and the application can be critical. Minor issues are:
the terms need to be consistent throughout the text body and should be precise. ie. PLA1 activity, PlaA, PlaA, ...etc.
Author Response
PLA1 is used for the general phospholipase A1, and PlaA is used for Serratia marcescens PLA1. The PlaA gene is represented as plaA. We have unified the usage of these throughout the manuscript.
In line 335, we have revised “fluorogenic substrate specific for PLA1” to “fluorogenic substrate specific for PlaA.” We have also revised “Activity of secreted PLA1” on the right y-axis of Figure 4A to “Activity of secreted PlaA.”
Reviewer 2 Report
The article from Park et al. entitled “High-level production of bacteriotoxic PLA1 in bacterial host P. fluorescens via ABC-transporter mediated secretion and inducible expression” deals with a new system to express phospholipase A1
My advice for this article is “Minor revisions”. The article is very interesting as it deals with a novel and efficient expression system in Pseudomonas to overexpress an enzyme that is very difficult to obtain in high amount. The experiments are elegantly designed and the results are particularly convincing. I particularly appreciate the sketch on Figure 1. This paper is an important breakthrough to obtain sufficient amount of PLA1 to allow its crystallization.
I have however 2 minor comments that have to be addressed by the authors and a flaw that has to be corrected:
Line 98 “we added a His-Tag on the 3’ end position of the plaA gene” is incorrect, a peptidic sequence cannot be added to a nucleotidic sequence. Please rephrase as “we added codons (or a sequence coding for) of a His-Tag on the 3’ end... plaA gene”. Line 102 “His-tagged plaA” is incorrect, please modify as “the sequence coding for His-tagged PlaA” Figure 2E. A proper control should be added to this figure to prove the specificity of the antibody (especially in the cell lysate, a sample from an ABC construct (id est without PlaA) must be run and tested). Anti-His antibodies are well known to recognize aspecifically endogenous E. coli proteins. To prove that PlaA is overexpressed in ABC-plA or in ABC/PlaA/PlaS extracts, an absence of signal should be obtained in ABC extract. The problem is the same on Figure 3C with proteins from P. fluorescens: a proper control (a protein extract coming from the ABC sample (without any PlaA or PlaS)) must be tested by western blot to prove the specificity of the antibody in Pseudomonas.
Author Respons
1. According to your comment, in line 98, we have revised the phrase “we added a His-tag on the 3’end position of the palA gene” to “we added codons of a His-Tag at the 3′ end…plaA gene.”
2. In line 102, we have revised “His-tagged plaA” to “the sequence coding for His-tagged PlaA.”
3. We did not present the results of SDS–PAGE with ABC extracts in Fig 2E and Fig 3C because we already knew that the target band of PlaA was not detected in the sample expressing only ABC transporter. We performed other experiments to show the specificity of His-antibody, data of which are not included in the main text; the results are attached below.
Accordingly, we have modified Supplementary Figure S2B to present the control vector pDSK519. This result shows that P. fluorescens does not express any band corresponding to PlaA without the plaA gene (lane 1):